# Repair Effect of Umbilical Cord Mesenchymal Stem Cells Embedded in Hydrogel on Mouse Insulinoma 6 Cells Injured by Streptozotocin

**DOI:** 10.3390/polym16131845

**Published:** 2024-06-28

**Authors:** Jia Yang, Yang Liu, Minghui Wang, Shengqin Chen, Qingya Miao, Zhicong Liu, Bin Zhang, Guodong Deng

**Affiliations:** Guangdong Provincial Key Laboratory of Marine Biotechnology, Department of Biology, College of Science, Shantou University, Shantou 515063, China; 20jyang3@stu.edu.cn (J.Y.); 21mhwang@stu.edu.cn (M.W.); 19sqchen@stu.edu.cn (S.C.); 21qymiao@stu.edu.cn (Q.M.); 17zcliu2@stu.edu.cn (Z.L.); 18bzhang3@stu.edu.cn (B.Z.); 20gddeng@stu.edu.cn (G.D.)

**Keywords:** umbilical cord mesenchymal stem cells, hydrogel, mouse insulinoma 6 cells injured by streptozotocin, oxidative stress, repair effect

## Abstract

Umbilical cord mesenchymal stem cells (UC-MSCs) possess the capabilities of differentiation and immune modulation, which endow them with therapeutic potential in the treatment of type 2 diabetes mellitus (T2DM). In this study, to investigate the repair mechanism of UC-MSCs in hydrogel on pancreatic β-cells in diabetes, mouse insulinoma 6 (MIN-6) cells damaged by streptozotocin (STZ) in vitro were used in co-culture with UC-MSCs in hydrogel (UC-MSCs + hydrogel). It was found that UC-MSCs + hydrogel had a significant repair effect on injured MIN-6 cells, which was better than the use of UC-MSCs alone (without hydrogel). After repair, the expression of superoxide dismutase (SOD) and catalase (CAT) as well as the total antioxidant capacity (T-AOC) of the repaired MIN-6 cells were increased, effectively reducing the oxidative stress caused by STZ. In addition, UC-MSCs + hydrogel were able to curb the inflammatory response by promoting the expression of anti-inflammatory factor IL-10 and reducing inflammatory factor IL-1β. In addition, the expression of both nuclear antigen Ki67 for cell proliferation and insulin-related genes such as *Pdx1* and *MafA* was increased in the repaired MIN-6 cells by UC-MSCs + hydrogel, suggesting that the repair effect promotes the proliferation of the injured MIN-6 cells. Compared with the use of UC-MSCs alone, UC-MSCs + hydrogel exhibit superior antioxidant stress resistance against injured MIN-6 cells, better proliferation effects and a longer survival time of UC-MSCs because the porous structure and hydrophilic properties of the hydrogel could affect the growth of cells and slow down their metabolic activities, resulting in a better repair effect on the injured MIN-6 cells.

## 1. Introduction

The pathogenesis of type 2 diabetes mellitus (T2DM) is characterized by inadequate β-cell function leading to insufficient insulin secretion. There is increasing evidence that pancreatic β-cell dysfunction is a key determinant of disease progression in T2DM [1]. Chronic, persistent inflammation is identified as an important factor causing the functional damage and apoptosis of islet β-cells [2]. Oxidative stress can cause cells to overproduce reactive oxygen species (ROS), which can lead to an increase in intracellular oxygen free radical concentrations and cause oxidative damage to proteins, lipids and DNA. These oxidatively modified biomolecules can damage the structure and function of cells, leading to the apoptosis and dysfunction of pancreatic islet β-cells [3,4,5,6]. In addition, oxidative stress can also impair insulin secretion and synthesis, further exacerbating the development of T2DM.

A number of studies have shown that umbilical cord mesenchymal stem cells (UC-MSCs) can secrete extracellular vesicles containing rich bioactive substances such as miRNA, proteins and mRNA, which can influence insulin secretion, β-cell proliferation and immune regulation through paracrine effects [7,8,9,10]. In addition, factors secreted by UC-MSCs can protect islet β-cells by inhibiting oxidative stress and reducing the apoptosis of islet β-cells. In addition, UC-MSCs can inhibit the release of inflammatory factors through paracrine action, improve islet inflammation and protect islet β-cell function [11,12,13,14,15]. Therefore, UC-MSCs are widely used in diabetes treatment. Previous studies have shown that UC-MSCs can stimulate the proliferation and regeneration of islet β-cells, improve insulin secretion and control blood glucose levels in diabetes [16,17]. UC-MSC therapy can promote the regeneration and functional recovery of β-cells, attenuate the dedifferentiation of β-cells caused by the inflammatory microenvironment in diabetes, and improve the ability to secrete insulin [18,19,20]. However, the direct injection method of UC-MSC therapy has some obvious drawbacks. The inflammatory and oxidative stress pathological microenvironment of T2DM may impair stem cell viability and reduce UC-MSCs’ therapeutic efficacy. Moreover, the instability of differentiation and the potential tumorigenicity of the direct injection method could elicit immune responses or lead to delayed toxicities, such as tumor formation and metastasis [21,22]. UC-MSCs, through direct injection, are difficult to accurately reach and remain completely in the damaged tissues, which also significantly affects their therapeutic efficacy.

To ensure that UC-MSCs effectively exert their function at the target site during UC-MSC therapy, it is necessary to develop a cell carrier that provides a good microenvironment for the embedding of UC-MSCs [23,24,25]. The polysaccharide-based hydrogel has a porous network structure with high water uptake, good biocompatibility and low immunogenicity or toxicity. It could be used as a biomimetic extracellular matrix (ECM) to create a suitable microenvironment for the growth and differentiation of UC-MSCs, so the composite hydrogel mimicking the ECM could become an effective UC-MSC carrier for T2DM treatment [26]. A composite hydrogel (CS/CSG/CO) composed of chitosan (CS), chitosan biguanide hydrochloride (CSG) and collagen (CO) has already been developed in our laboratory, which has excellent cell compatibility. It can be used to investigate the repair effect of UC-MSCs embedded in the composite hydrogel on injured islet β-cells.

The mouse insulinoma 6 (MIN-6) cell line was originally isolated from a mouse pancreas. MIN-6 has become an important islet model β-cells for the study of insulin secretion and regulation and is frequently used in studies of T2DM [27]. Therefore, in the present study, the effect of UC-MSCs embedded in CS/CSG/CO on MIN-6 cells damaged by STZ is investigated and the repair mechanism is preliminarily explored.

## 2. Materials and Methods

### 2.1. Materials

Chitosan (CS, degree of deacetylation ≥ 95%, Mw = 300 kDa) was purchased from Prius Bioengineering Co. (Xi’an, China). Sodium β-glycerophosphate (GP) was purchased from Sigma-Aldrich (Shanghai, China). Sodium bicarbonate was purchased from Guanghua Technology Co. (Guangdong, Shantou, China). Sterile phosphate buffered solution (PBS), fetal bovine serum (FBS) and Dulbecco’s modified Eagle medium (DMEM) were purchased from Gibco Co. (New York, NY, USA). Penicillin–streptomycin, trypsin (0.25 *w*/*v*% trypsin with phenol red), calcein/PI cell activity and cytotoxicity assay kits were purchased from Beyotime Biotech Inc. (Shanghai, China). Fish skin collagen (CO, purity ≥ 90%, Mw = 3–5 kDa), the malondialdehyde (MDA) kit, and the lactate dehydrogenase (LDH) kit were purchased from Solarbio Science & Technology Co. (Beijing, China). The superoxide dismutase (SOD) kit, total antioxidant capacity (T-AOC) kit, catalase (CAT) kit, inducible nitric oxide synthase (iNOS) kit, interleukin-1β (IL-1β) kit, interleukin-10 (IL-10) kit, nuclear antigen Ki67 (Ki67) kit, pancreatic and duodenal homeobox 1(*pdx1*) kit, musicoaponeurotic fibrosarcoma oncogene homolog A (*MafA*) kit, and insulin kit were obtained from Andy Gene Co. (Shanghai, China). Mouse UC-MSCs and mouse MIN-6 cells were purchased from Xuanya Biotechnology Co. (Shanghai, China). Complete culture medium for mouse UC-MSCs and complete culture medium for mouse MIN-6 cells were purchased from Procell Life Science & Technology Co. (Wuhan, China).

### 2.2. Cells Culture

MIN-6 and UC-MSCs were cultured in Dulbecco’s modified Eagle medium (DMEM) with high glucose concentrations and 15% fetal bovine serum at 37 °C in a cell incubator with 5% CO_2_. After digestion with 0.25% trypsin for 1 min, the MIN-6 cells were subcultured in different flasks and subcultured every 2–3 days. The number of both cell passages did not exceed six generations.

The UC-MSCs embedded in the hydrogel were cultured according to specific steps. The raw materials of the hydrogel, including CS, CSG and CO, were sterilized with a 0.2 μm membrane and then used to prepare the composite hydrogel. On the basis of physical crosslinking, the CS/CSG/CO composite hydrogel was prepared from 2 wt% CS, 50 wt% GP, 2 wt% CSG and 0.5 wt% CO. The specific preparation method consisted of mixing 6 mL of CSG, 3 mL of CS, 3 mL GP and 1 mL of CO, adjusting the pH with sodium bicarbonate (NC) and then forming a gel at 37 °C. The UC-MSCs concentration was adjusted to 1 × 10^6^ cells/mL, then they were added to the hydrogel precursor liquid and mixed uniformly. The precipitated UC-MSCs were suspended in the sol, placed in a perforated plate and incubated to form a gel. After gel formation, a fresh culture medium was added.

### 2.3. Experimental Grouping

Streptozocin (STZ) was dissolved in DMEM to prepare a 3 mmol/L STZ culture medium. MIN-6 cells were cultured in 3 mmol/L STZ culture medium for 12 h, digested with 0.25% trypsin for 1 min and centrifuged for later use.

The all MIN-6 cells concentration was initially adjusted to 1 × 10^6^ cells/mL, then they were inoculated as follows. The normal MIN-6 cells were inoculated into the well plates as the normal control group, and the MIN-6 cells treated with STZ for 12 h were inoculated into the well plates as the negative control group. The other two experimental groups included MIN-6 cells treated with STZ and cultured with an equal volume of UC-MSCs in DMEM for 24 h and MIN-6 cells treated with STZ and cultured with an equal volume of UC-MSCs embedded in hydrogel. The concentration of UC-MSCs in the two experimental groups was adjusted to 1 × 10^6^ cells/mL. Thus, the cultured MIN-6 cells were divided into four groups with 5 wells each, including the control group (in DMEM only), the STZ injury group (STZ), the UC-MSCs repair group (STZ + UC-MSCs in DMEM) and the UC-MSCs + hydrogel repair group (STZ + UC-MSCs embedded in hydrogel).

### 2.4. Repairing Effect of UC-MSCs in Combination with Hydrogel on STZ-Induced Injury in MIN-6 Cells

To demonstrate the repair effect of UC-MSCs + hydrogel on damaged MIN-6 cells, the following indicators were measured after the 24 h co-culture of UC-MSCs with MIN-6 cells.

#### 2.4.1. MDA and LDH Determination

After 24 h of cultivation, cells were trypsinized as previously described and MDA and LDH concentrations were quantified using a colorimetric method [28]. All procedures were performed in strict compliance with the reagent kit instructions.

#### 2.4.2. SOD, T-AOC, CAT Determination

The cell concentration was set to 1 × 10^6^ cells/mL as described in Section 2.3. The MIN-6 cell suspensions were randomly divided into a normal control group, a negative control group, a UC-MSC repair group and a UC-MSCs + hydrogel repair group. After the 24 h culture, the supernatant was collected and the SOD content [3], T-AOC [29] and CAT content [30] were determined by radioimmunoassay according to the instructions of the kit.

#### 2.4.3. Inducible Nitric Oxide Synthase (iNOS), IL-1β, IL-10 Assay

Cell suspensions were prepared using the above method and then randomly assigned to the normal control group, the negative control group, the UC-MSC repair group and the UC-MSCs + hydrogel repair group. The cells were then inoculated into a 96-well plate and incubated in a CO_2_ incubator for 24 h. After culturing, the cell supernatant was collected and the levels of iNOS, IL-1β and IL-10 were determined by radioimmunoassay according to the kit instructions [31,32,33].

#### 2.4.4. Ki67, *Pdx1*, *MafA* Assay

Cell suspensions were prepared by the above method and randomly assigned to the normal control group, the negative control group, the UC-MSC repair group and the UC-MSCs + hydrogel repair group. Each group was cultured for 24 h and the cell supernatant was collected. The levels of Ki67, *Pdx1* and *MafA* in the supernatant were determined by radioimmunoassay [34,35]. All steps were performed strictly according to the instructions of the reagent kit.

#### 2.4.5. Insulin Measurement

Cell suspensions were prepared by the above method and randomly assigned to the normal control group, the negative control group, the UC-MSC repair group and the UC-MSCs + hydrogel repair group. After each group had been cultured for 24 h, the culture medium was aspirated and washed once with phosphate buffer (PBS). High-glucose DMEM culture medium without serum was then added and the cells were cultured for 12 h. The cell supernatant was then collected and insulin release in the cell supernatant was measured using a radioimmunoassay [36].

#### 2.4.6. Determination of Glucose Content in Cell Supernatant

The cells were randomly divided into the normal control group, the negative control group, the UC-MSC repair group, the UC-MSCs + hydrogel repair group and the CSG group. After 24 h of cultivation, the culture medium was aspirated, washed once with PBS and cultured for 24 h in DMEM medium without serum. The supernatant was collected and the glucose content in the supernatant was determined. All steps were performed strictly according to the kit instructions [3].

#### 2.4.7. Live and Dead Cell Staining

To further investigate the effects of UC-MSCs + hydrogel on MIN-6 cells, the cells were randomly divided into the group with UC-MSCs and the group with UC-MSCs + hydrogel, which repaired the cells. After 4, 8, 12, 14, 16, 20, 24, 28 and 32 days of cultivation, the live and dead cells were stained according to the instructions of the kit. All procedures strictly followed the method adapted from Bhatt et al.’s study [37]. The Calcein AM was used, which is an excellent fluorescent dye for live cell labeling, capable of readily penetrating the cell membrane of viable cells to generate a green, fluorescent substance with a strong fluorescence signal. In contrast, for dead cells with compromised cell membranes, propidium iodide (PI) was used, which can enter the cell, bind to nucleic acids, and produce a bright red fluorescent signal indicative of dead cells. A quantitative assessment of cell viability was performed using the image processing software Image J 1.51 j 8 for the quantitative analysis of fluorescence intensity. The co-culture procedure of UC-MSCs with MIN-6 cells was as follows. The composite hydrogel precursor solution was prepared at 25 °C and mixed with the counted UC-MSCs, and then we let the gel precursor solution sit to form a gel at 37 °C. After the gel had formed, the MIN-6 cells were seeded onto the surface of the gel, then cultured with the addition of growth medium.

### 2.5. Statistical Significance

The statistical analyzes, including the ANOVA analysis and Dunnett’s multiple comparison test, were performed using SPSS 22.0, Professional Edition software. A *p*-value below 0.05 was considered statistically significant (* *p* < 0.05, ** *p* < 0.005, *** *p* < 0.001). The experimental data were expressed as mean ± standard deviation.

## 3. Results and Discussion

### 3.1. UC-MSCs + Hydrogel Reduced Oxidative Stress of Injured MIN-6 Cells

STZ treatment induces the production of a large amount of ROS in MIN-6 cells, which leads to oxidative stress and causes significant damage to the DNA, proteins and lipids of the cells. An excess of ROS induces lipid peroxidation and forms lipid peroxides, whose metabolite MDA can serve as an indicator of the strength of lipid peroxidation reactions [38]. Figure 1 displays the oxidative stress in the injured MIN-6 cells of various experimental groups.

Compared to the normal control group, MDA increased in the supernatant of the STZ injury group, the UC-MSC repair group and the UC-MSCs + hydrogel repair group (Figure 1a). In addition, the STZ injury group showed higher MDA values compared to the UC-MSCs + hydrogel repair group, while the UC-MSCs + hydrogel repair group showed a significant decrease in MDA compared to the UC-MSC repair group (*p* < 0.05). These results indicate that the combined treatment of UC-MSCs + hydrogel can partially alleviate the cell damage induced by STZ treatment, and the repair effect was superior to that of the UC-MSC repair group alone. Oxidative stress on cells leads to oxidation and the damage of lipid components, proteins, nucleic acids and other structures of the cell membrane, resulting in increased permeability and impaired membrane integrity. The release of LDH was closely related to the disruption of cell membrane integrity [3]. The result (Figure 1b) showed that the LDH content in the supernatant of the UC-MSCs + hydrogel repair group was lower than that of the other three groups. In addition, the LDH content was significantly higher in the STZ injury group and the UC-MSC repair group compared to the UC-MSCs + hydrogel repair group (*p* < 0.001). It can be concluded that treatment with UC-MSCs + hydrogel can significantly reduce the cell damage caused by STZ. The enhanced therapeutic effects may be attributed to the protective microenvironment for UC-MSC growth provided by the composite hydrogel [39,40]. UC-MSCs embedded in hydrogel could exert their paracrine functions more effectively and facilitate the repair of injured MIN-6 cells. The protection and functional enhancement for UC-MSCs are responsible for the improved therapeutic effects of the UC-MSCs + hydrogel repair group [7,11,15].

### 3.2. UC-MSCs + Hydrogel Enhanced Antioxidant Capacity of Injured MIN-6 Cells

Figure 2 displays the different antioxidation evaluations in the injured MIN-6 cells of various experimental groups. STZ treatment led to a decrease in the SOD content. STZ can trigger oxidative stress in the cells and thus reduce the SOD content. The results (Figure 2a) showed a significant decrease in SOD content in the STZ-damaged group compared to the normal control group (*p* < 0.01). In addition, the SOD content was lower in the STZ-injured group than in the UC-MSC repair group and the UC-MSCs + hydrogel repair group. In addition, the SOD content was significantly lower in the STZ injury group than in the UC-MSCs + hydrogel repair group (*p* < 0.05). It can be concluded that UC-MSCs + hydrogel treatment significantly attenuated the cell injury caused by STZ-induced oxidative stress [41].

T-AOC stands for the antioxidant capacity of cells and reflects their cellular antioxidant capacity. The experimental results showed a decrease in T-AOC in the STZ-damaged group, which was the lowest among the four groups (Figure 2b). In addition, the T-AOC of the UC-MSC repair group showed a significant improvement compared with the STZ-injured group (*p* < 0.05), and the T-AOC of the UC-MSCs + hydrogel repair group also showed a significant increase (*p* < 0.01). These results support the role of UC-MSCs + hydrogel in reducing oxidative stress in cells and improving the ability of MIN-6 cells to respond to oxidative stress [29].

CAT activity may reflect the response of cells to oxidative stress and their ability to maintain redox balance. The main function of CAT is the decomposition of hydrogen peroxide (H_2_O_2_) into water and oxygen. Since hydrogen peroxide is a harmful oxidizing agent that can damage cell structure and function, CAT plays a crucial role in protecting cells from oxidative stress [29]. The experimental results (Figure 2c) showed that CAT expression in the normal control group was the highest among the four groups, while CAT expression in the STZ-damaged group was lower than in the other three groups. The CAT expression level in the group with UC-MSCs + hydrogel repair was higher than that in the STZ-injured group and approached the level of the normal control group. These results suggest that UC-MSCs + hydrogel may help cells reduce the concentration of harmful H_2_O_2_ and protect them from oxidative stress damage. Moreover, for the UC-MSCs + hydrogel group, not only the paracrine actions of UC-MSCs but also the modified chitosan guanidine hydrochloride (CSG) in the hydrogel could contribute to the improved antioxidant capacity [7,8]. CSG possesses inherent antioxidant stress capabilities [42,43,44], which could further augment the antioxidant capacity of the UC-MSCs + hydrogel group [45,46].

### 3.3. UC-MSCs + Hydrogel Reduced Inflammation of Injured MIN-6 Cells

Figure 3 displays the inflammation (iNOS and IL-1β) and anti-inflammation (IL-10) cytokines in the injured MIN-6 cells of various experimental groups. In oxidative stress, iNOS is involved in the inflammatory and cell damage processes triggered by oxidative stress. The degree of iNOS activation may reflect the extent of cellular damage [47,48,49]. The experimental results (Figure 3a) showed that the expression level of iNOS was lower in the normal control group. In contrast, it was significantly higher in the STZ-damaged group than in the other three groups (*p* < 0.001). Of note, the expression of iNOS was significantly lower in the UC-MSCs + hydrogel repair group than in the STZ-injured group and approached the level of the normal control group. These results suggest a reparative effect of UC-MSCs + hydrogel on STZ-induced MIN-6 cell injury.

IL-1β is a pro-inflammatory cytokine that plays a crucial role in immune response and inflammation. The experimental results (Figure 3b) showed that the expression level of IL-1β was lower in the normal control group, higher in the STZ injury group than in the other three groups, and significantly higher than in the UC-MSC repair group and UC-MSCs + hydrogel repair group (*p* < 0.01). In addition, the expression level of IL-1β in the UC-MSCs + hydrogel repair group was significantly lower than that in the STZ injury group and approached the level of the normal control group. These results suggest that UC-MSCs + hydrogel may have some reparative effect on the inflammation caused by STZ-induced MIN-6 cell injury [50].

IL-10 has anti-inflammatory and immunosuppressive properties that can inhibit the production of cytokines, pro-inflammatory proteins and chemokines, thereby alleviating the inflammatory process. It plays an important role in maintaining the balance of the immune system and inhibiting an excessive inflammatory response. As shown in Figure 3c, the normal control group had lower IL-10 expression, while the STZ-damaged group had significantly lower IL-10 expression than the other three groups (*p* < 0.05). The UC-MSC repair group and the UC-MSCs + hydrogel repair group had significantly higher levels of IL-10 expression than the STZ-injured group (*p* < 0.05), approaching the level of the normal control group. These results suggest that UC-MSCs + hydrogel have some inhibitory effect on the inflammation induced by STZ-induced MIN-6 cell injury and have some immunoregulatory ability [51,52]. The UC-MSCs + hydrogel group demonstrated favorable anti-inflammatory properties, which could be also attributed to the above two main factors, the paracrine effects of UC-MSCs [8,9] and the incorporation of CSG with the intrinsic anti-inflammatory property [49].

### 3.4. UC-MSCs + Hydrogel Promoted Injured MIN-6 Cells Proliferation

Figure 4 displays the different expression levels of cell proliferation markers in the injured MIN-6 cells of various experimental groups. Ki67 is a cell proliferation marker whose expression level is higher in actively proliferating cells. In islet cells, the expression level of Ki67 may reflect the proliferation activity of islet β-cells [53,54]. When the expression level of Ki67 in the cells was determined (Figure 4a), it was found that the expression level of Ki67 decreased significantly in the STZ-damaged group compared to the normal control group (*p* < 0.05). In contrast, the expression of Ki67 significantly increased in the group with UC-MSCs + hydrogel (*p* < 0.05), suggesting that UC-MSCs + hydrogel can protect MIN-6 cells from STZ injury and improve their function. *Pdx1* is an important transcription factor of islet cells, which is crucial for islet cell development and function. *Pdx1* promotes the differentiation of islet precursor cells into islet cells during early embryonic pancreatic development and continues to regulate islet cell maturation and function at subsequent stages of development, specifically promoting islet β-cell maturation and insulin secretion [53,54]. The expression of *Pdx1* may reflect the functional status of β-cells, as observed in our study, where the expression of *Pdx1* (Figure 4b) was significantly downregulated in the STZ injury group compared to the normal control group (*p* < 0.05). Conversely, the expression of *Pdx1* was increased in the UC-MSC repair group and the UC-MSCs + hydrogel repair group, suggesting that UC-MSCs + hydrogel significantly increased the proliferation activity of MIN-6 cells. *MafA* plays an important role in pancreatic islet cell differentiation and function. *MafA* is also involved in the migration of pancreatic progenitor cells to the maturation process of β-cells. It is involved in the regulation of insulin gene transcription and promotes insulin synthesis and secretion. *MafA* is involved in the stress response of islet cells and is an important regulator of the stress response of islet β-cells to hyperglycemia and insulin resistance [55]. Based on the Ki67 and *Pdx1* results, we detected the expression of *MafA* in MIN-6 cells as shown in the figure (Figure 4c). The results showed that the expression of *MafA* was higher in the UC-MSC repair group and the UC-MSCs + hydrogel repair group compared with the STZ injury group, suggesting that UC-MSCs + hydrogel can repair the function of MIN-6 cells and increase their proliferation activity. The composite hydrogel facilitated the optimal paracrine action of UC-MSCs and enhanced the proliferation of MIN-6 cells, which held significant implications for the repair of β-cells in T2DM [8,9].

### 3.5. UC-MSCs + Hydrogel Enhanced Biological Function of Injured MIN-6 Cells

Insulin is the only hormone in the body that lowers the blood glucose concentration and thus contributes to the maintenance of a normal blood glucose level and the energy supply of the cells. Figure 5 displays the insulin concentration and glucose concentration in injured MIN-6 cells of various experimental groups. The results (Figure 5a) showed that the insulin secretion of the STZ-damaged group was significantly lower than that of the UC-MSC-repaired group and the UC-MSCs + hydrogel group (*p* < 0.05). These results suggest that UC-MSCs + hydrogel could restore the function of β-cells and improve insulin secretion from MIN-6 cells [56,57]. It has been postulated that UC-MSCs + hydrogel have the ability to promote β-cell proliferation.

Determining the glucose concentration in the cell supernatant is important for understanding the regulatory mechanism of insulin secretion. β-cells are responsible for insulin secretion, and changes in glucose concentrations have a direct effect on insulin secretion. Impaired β-cells can lead to reduced or stopped insulin secretion, resulting in the reduced utilization of glucose by the cells [3]. As shown in Figure 5b, the glucose content in the supernatant of the STZ-impaired group was the highest among the five groups (*p* < 0.05), indicating decreased glucose utilization due to impaired MIN-6 cells and the development of insulin resistance. In addition, glucose levels were significantly lower in the UC-MSC repair group and the UC-MSCs + hydrogel repair group than in the STZ injury group (*p* < 0.001). In addition, the glucose content was found to be significantly lower in the UC-MSCs + hydrogel repair group and the UC-MSCs + CSG repair group than in the STZ injury group (*p* < 0.001). This result suggests that UC-MSCs + hydrogel have the ability to repair the function of MIN-6 cells and improve the insulin secretion of MIN-6 cells. It also showed that CSG has a repairing effect on impaired MIN-6 cells and improves the glucose sensitivity of MIN-6 cells.

### 3.6. UC-MSCs + Hydrogel Enhanced Long-Term Survival of Injured MIN-6 Cells

To evaluate the effect of UC-MSCs + hydrogel on the survival of MIN-6 cells, the staining of live and dead cell was performed in this study. Figure 6 shows that a large number of cells survived in the UC-MSC repair group during the 32-day experiment. However, the number of dead cells was higher than in the UC-MSCs + hydrogel repair group. Moreover, a considerable number of UC-MSCs survived within the hydrogel of the UC-MSCs + hydrogel repair group, and the MIN-6 cells co-cultured with UC-MSCs + hydrogel also showed a high survival rate. The UC-MSCs + hydrogel repair group showed less cell death compared to the UC-MSC repair group. In addition, the UC-MSCs + hydrogel repair group did not exhibit high cell density, leading to cell floating, which was probably due to the encapsulation effect of the hydrogel on UC-MSCs. The adequate porosity of the hydrogel provides more space for cell survival [58,59]. There was a significant difference between the UC-MSC repair group and the UC-MSCs + hydrogel repair group (*p* < 0.001), and the interaction effect was also statistically significant (*p* < 0.001). This suggests that the approach of using stem cells encapsulated in hydrogels in combination with T2DM may reduce the need for stem cells and mitigate the risk of tumorigenicity associated with stem cells.

In this experiment, the long-term survival of UC-MSCs in hydrogel was observed and found to be important to reduce the number of injections and avoid delayed toxicity due to stem cell retention [21]. UC-MSCs + hydrogel have significant advantages over existing non-hydrogel-encapsulated stem cell approaches in diabetes treatment. The above experimental results show that UC-MSCs + hydrogel can reduce oxidative stress and inflammation, promote MIN-6 cell proliferation, and restore the biological functions of MIN-6 [16,17,18]. In addition, current research in the field of non-hydrogel-encapsulated stem cells for the treatment of diabetes shows that the direct injection of stem cells can reduce inflammation, modulate immune responses and promote cell function. However, this study with UC-MSCs + hydrogel showed the prolonged survival and sustained therapeutic effect of the cells in the hydrogel, which is not usually observed with direct injection [21]. Existing methods often involve the direct injection of stem cells for the treatment of diabetes. However, this approach may not provide the same protection and support for the cells as embedding them in hydrogel nor the same sustained therapeutic effect as hydrogel co-culture [17,18].

## 4. Conclusions

UC-MSCs + hydrogel showed the ability to repair the toxic effects of STZ on MIN-6 cells, as evidenced by the reduction in the release of MDA and LDH, the increase in the release of SOD and CAT, the modulation of oxidative stress and anti-inflammatory effects, and the enhancement of β-cell proliferation in MIN-6 cells, as well as the increased expression of Ki67, *Pdx1* and *MafA*. In addition, UC-MSCs + hydrogel was found to enhance insulin release and glucose metabolism in MIN-6 cells, contributing to improved glucose utilization. The 32-day staining experiment with live and dead cells confirmed the prolonged survival of UC-MSCs in the hydrogel, such that their extended co-cultivation with MIN-6 cells could effectively support the repair of injured MIN-6 cells. These results suggest that UC-MSCs + hydrogel can reduce STZ-induced oxidative stress and cellular inflammation through a paracrine effect. Moreover, UC-MSCs + hydrogel can reverse β-cell dedifferentiation and improve β-cell function as well as increase insulin resistance and the glucose utilization rate in MIN-6 cells.

Based on the above study, the interactions between the composite hydrogel and the stem cells was necessary to be deeply explored in their therapeutic potential for diabetes treatment. The development of large-scale stem cell applications, standardized preparation and the establishment of therapeutic evaluation systems were crucial to ensure the safety and efficacy of clinical applications. Additionally, the influence of the T2DM microenvironment on the therapeutic potential of UC-MSCs and the long-term stability and function of the treated cells post treatment were key areas for future application. These studies should be prepared to offer novel strategies and insights into stem cell treatments for T2DM.

## Figures and Tables

**Figure 1 polymers-16-01845-f001:**
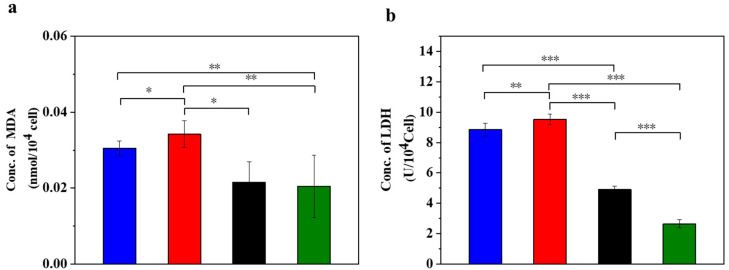
UC-MSCs + hydrogel reduced oxidative stress in injured MIN-6 cells. (**a**) MDA release level detection. (**b**) LDH release level detection. From left to right, the groups are con group (blue column), T2DM group (red column), UC-MSC group (black column), UC-MSCs + hydrogel group (green column). The intergroup comparison was applied by one-way analysis of variance (ANOVA), in which asterisks indicated the difference from the control group (* *p* < 0.05, ** *p* < 0.01, *** *p* < 0.001).

**Figure 2 polymers-16-01845-f002:**
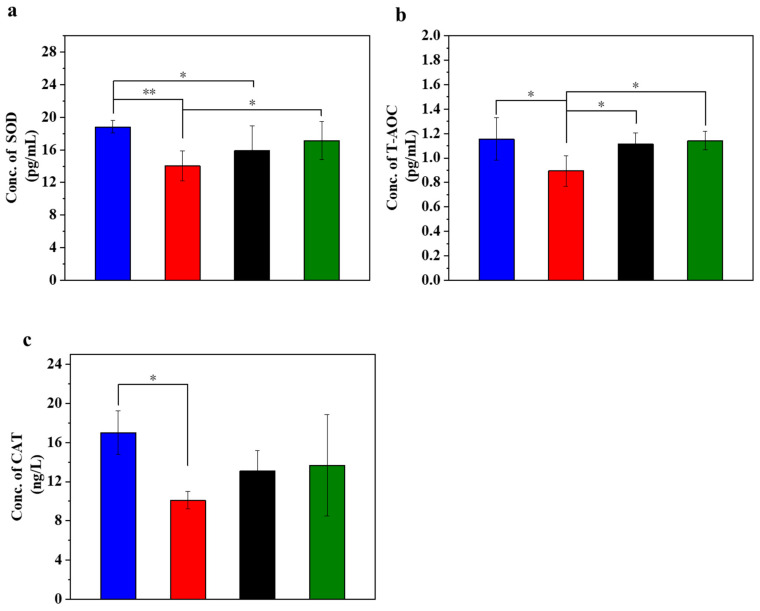
UC-MSCs + hydrogel enhanced antioxidant capacity of injured MIN-6 cells. (**a**) SOD expression level was detected. (**b**) T-AOC testing. (**c**) Detection of CAT expression level. From left to right, the groups are con group (blue column), T2DM group (red column), UC-MSC group (black column), UC-MSCs + hydrogel group (green column). The intergroup comparison was applied by one-way analysis of variance (ANOVA), in which asterisks indicated the difference from the control group (* *p* < 0.05, ** *p* < 0.01).

**Figure 3 polymers-16-01845-f003:**
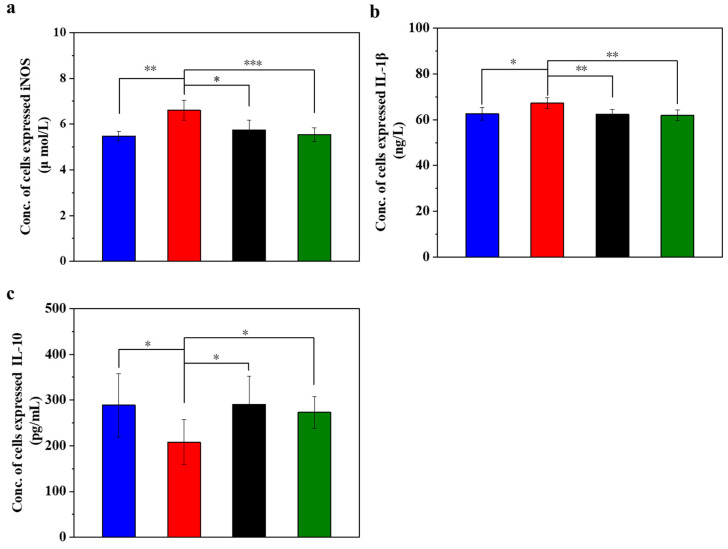
UC-MSCs + hydrogel reduced inflammation of injured MIN-6 cells. (**a**) Detection of iNOS expression level. (**b**) IL-1β testing. (**c**) IL-10 testing. From left to right, the groups are con group (blue column), T2DM group (red column), UC-MSC group (black column), UC-MSCs + hydrogel group (green column). The intergroup comparison was applied by one-way analysis of variance (ANOVA), in which asterisks indicated the difference from the control group (* *p* < 0.05, ** *p* < 0.01, *** *p* < 0.001).

**Figure 4 polymers-16-01845-f004:**
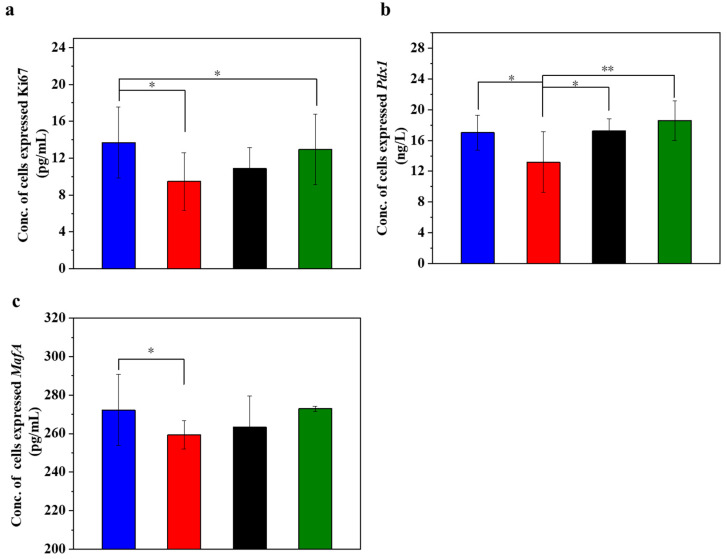
UC-MSCs + hydrogel promoted injured MIN-6 cells proliferation. (**a**) Ki67 expression level was detected. (**b**) *Pdx1* expression level detection. (**c**) *MafA* expression level detection. From left to right, the groups are con group (blue column), T2DM group (red column), UC-MSC group (black column), UC-MSCs + hydrogel group (green column). The intergroup comparison was applied by one-way analysis of variance (ANOVA), in which asterisks indicated the difference from the control group (* *p* < 0.05, ** *p* < 0.01).

**Figure 5 polymers-16-01845-f005:**
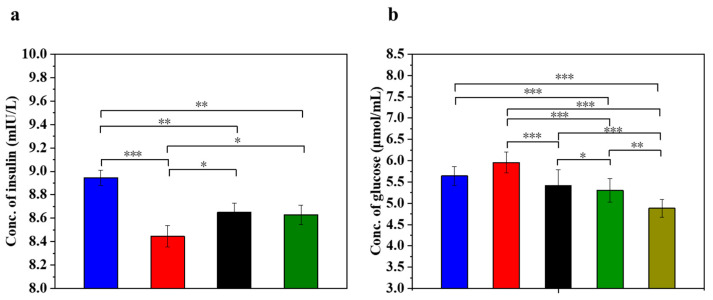
UC-MSCs + hydrogel enhanced biological function of injured MIN-6 cells. (**a**) Insulin expression levels were measured. (**b**) Determination of glucose content. From left to right, the groups are con group (blue column), T2DM group (red column), UC-MSC group (black column), UC-MSCs + hydrogel group (green column), UC-MSCs + CSG group (yellow-green column). The intergroup comparison was applied by one-way analysis of variance (ANOVA), in which asterisks indicated the difference from the control group (* *p* < 0.05, ** *p* < 0.01, *** *p* < 0.001).

**Figure 6 polymers-16-01845-f006:**
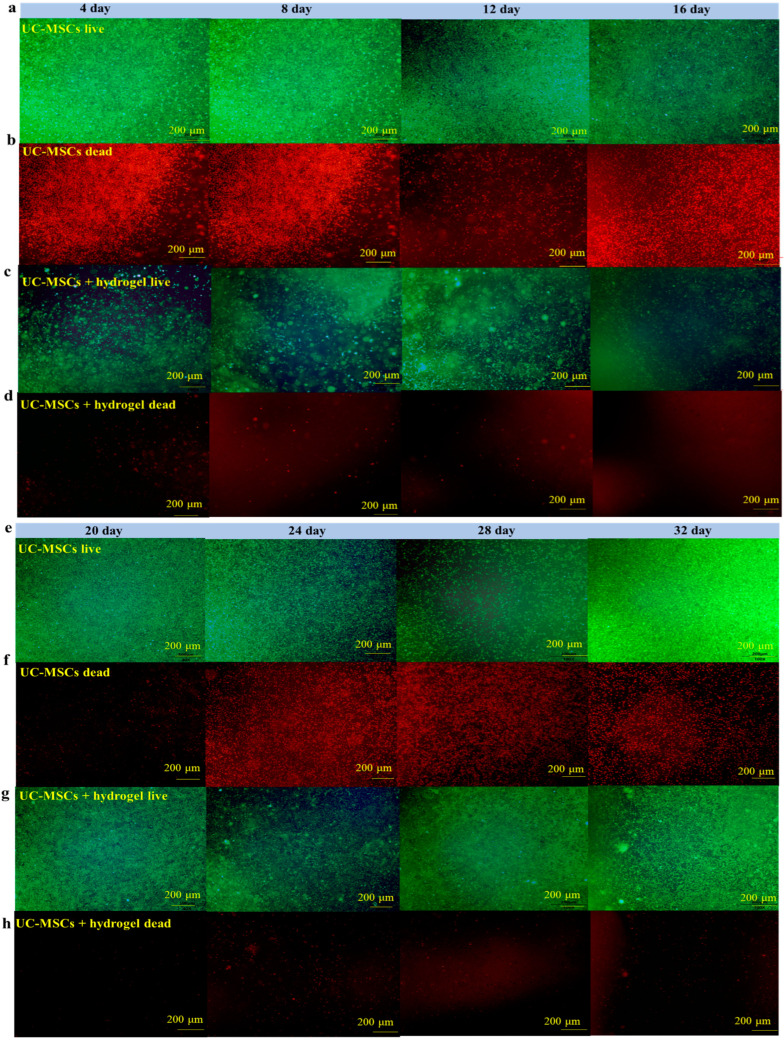
UC-MSCs + hydrogel enhanced long-term survival of injured MIN-6 cells. (**a**,**b**) UC-MSCs + MIN-6 group live cell staining and dead cell staining. (**c**,**d**) UC-MSCs + MIN-6 + hydrogel group live cell staining and dead cell staining. (**e**,**f**) UC-MSCs + MIN-6 group live cell staining and dead cell staining. (**g**,**h**) UC-MSCs + MIN-6 + hydrogel group live cell staining and dead cell staining. (**i**) Dead cell count. (**j**) Living cell count. The intergroup comparison was applied by one-way analysis of variance (ANOVA), in which asterisks indicated the difference from the control group (*** *p* < 0.001).

## Data Availability

The original contributions presented in the study are included in the article, further inquiries can be directed to the corresponding author.

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
