# Peer review of "Repair Effect of Umbilical Cord Mesenchymal Stem Cells Embedded in Hydrogel on Mouse Insulinoma 6 Cells Injured by Streptozotocin"

_polymers, 2024, doi:10.3390/polym16131845_

Round 1
Reviewer 1 Report
Comments and Suggestions for Authors
The present work is suitable for Polymer journal. However, major experimentation and revision is required before the acceptance of this manuscript.
Specific comments:
1. Please avoid abbreviations in the title for better understanding
2. Revised the abstract, including the background of the work, need, and study design
3. Keywords can be improved, avoid abbreviations
4. Check the whole manuscript for typo error
5. Line 66-70 Improve the content
6. Please check for abbreviation consistency
7. Authors can provide the SEM characterizations for hydrogel
8. Further, what about the viscosity and pH of the developed hydrogel
9. In figure captions, please include the statistical significance for the comparison
10. Include the future perspective of the study and limitations
Comments on the Quality of English LanguageMinor editing of English language required
Author Response
Comments 1. Please avoid abbreviations in the title for better understanding
Response: Thank you for the careful question. We have modified in the main text.
Comments 2. Revised the abstract, including the background of the work, need, and study design
Response: Thank you for the valuable question. We have modified the abstract in the revised manuscript.
Comments 3. Keywords can be improved, avoid abbreviations
Response: We have modified the keywords in the revised manuscript.
Comments 4. Check the whole manuscript for typo error
Response: Thank you for the careful question. We have modified in the main text.
Comments 5. Line 66-70 Improve the content
Response: Thank you for the valuable question. We have modified the line 68-75 in the revised manuscript.
Comments 6. Please check for abbreviation consistency
Response: Thank you for the careful question. We have modified in the main text.
Comments 7. Authors can provide the SEM characterizations for hydrogel
Response: In our previous studies, various physicochemical characterization information on composite hydrogels has been reported. The SEM characterizations for hydrogel have been provided in our previous paper “Thermosensitive and injectable hydrogel composed of modified chitosan embedding UC-MSCs for T2DM treatment” as following, which is displayed in fig.(a). Gel-1, Gel-2 and Gel-3 had different components concentration. Gel-
3 was used in this paper.
Figures: (a) SEM images of the composite hydrogels.
Comments 8. Further, what about the viscosity and pH of the developed hydrogel
Response: The viscosity of this composite hydrogel has not been tested, while the rheology test results of the three hydrogels at different temperatures were shown in the following Fig.(e). For the three hydrogels, the loss moduli (G") were generally lower than the storage moduli (G') under a certain temperature, indicating that the composite hydrogels were liquid state. When the temperatures were over the certain temperature, the curves of G' and G" intersected, which indicated that the hydrogels underwent a sol-gel transition. With further increasing temperature, G' became much larger than G", which indicated that the hydrogels exhibited solid properties with forming a gel state. The phase inversion temperatures of Gel-1, Gel-2 and Gel-3 were around at 42 °C, 37 °C and 34 °C, respectively. The results displayed that Gel-2 and Gel-3 were more suitable for the applications in physiological conditions than Gel-1.
The rheological properties of the three hydrogels were further determined by sweeping the frequency from 0.1 to 10 Hz as shown in Fig.(f). The three composite hydrogels had significantly greater G' values than G" values, which indicated that the polysaccharide-based hydrogels were elastic and solid state. Moreover, it was pointed out that Gel-1 had the lowest G' and G" values among the three hydrogels, the curves of G' and G" intersected with increasing frequency. It was hypothesized that Gel-1 only composed CS would form a semi-dilute solution as frequency increasing, yet Gel-2 and Gel-3 both with more complex compositions would not form the semi-dilute solution within the investigated frequency range. Moreover, the pH value of the composite hydrogel is about 7.0, which is close to the physiological pH.
Figures: (e) Effect of temperature on storage modulus (G') and loss modulus (G'') of the composite hydrogels. (f) Effect of frequency on storage modulus (G') and loss modulus (G'') of the composite hydrogel.
Comments 9. In figure captions, please include the statistical significance for the comparison
Response: Thank you for the careful question. It has been annotated below the figure captions in the revised manuscript.
Comments 10. Include the future perspective of the study and limitations
Response: Thank you for the valuable question. We have added the contents in line 450-458 of the revised manuscript.

Reviewer 2 Report
Comments and Suggestions for Authors
A very good article has been presented by the respected authors. There are several important points that must be resolved before any decision:
As for the abstract, authors should write a suitable conclusion section for it.
The sources used in the introduction are few. Also, newer and more relevant sources are needed.
Cells were in which passage number?
Statistical evaluations are incomplete. Please specify which test is used for which variable.
The quality of the images is not good.
The information of the figures is not fully indicated below the figure.
The discussion section could be better and with better details.
Author Response
Reviewer 2
Comments 1:A very good article has been presented by the respected authors. There are several important points that must be resolved before any decision:
As for the abstract, authors should write a suitable conclusion section for it.
Response: The general statement in line 20-22 has already been added in the abstract of the revised manuscript.
Comments 2:The sources used in the introduction are few. Also, newer and more relevant sources are needed.
Response: Thank you for the valuable question. We have added the new relevant literatures have been added in the introduction of the revised manuscript.
Comments 3:Cells were in which passage number?
Response: Thank you for the careful question. The number of cell passages does not exceed six generations, which has been added in the revised manuscript.
Comments 4:Statistical evaluations are incomplete. Please specify which test is used for which variable. The quality of the images is not good. The information of the figures is not fully indicated below the figure.
Response: Thank you for the valuable question. The statistical evaluations have been revised and added in the figure captions. The significant difference between different groups has been also indicated. We have raised and adjusted the resolution of the figures. We also carefully checked the information of the figures corresponding with the following figures.
Comments 5:The discussion section could be better and with better details.
Response: Thank you for the valuable question. We have modified the discussion sections in the revised manuscript.

Reviewer 3 Report
Comments and Suggestions for Authors
Dear Authors,
I have completed the review of your manuscript titled "Repair effect of UC-MSCs embedded in hydrogel on MIN-6 cells injured by STZ". However, I would like to recommend some revisions before further consideration.
1) What are the key findings of this study that contribute to the current understanding of diabetes treatment? How does this research advance the field and add to the existing knowledge?
2) Please provide more information on the statistical methods used to compare the different experimental groups and determine the significance of the results. Were the statistical tests appropriate for the data analysis?
3) It would be helpful to have a group with STZ-induced injury but without any repair treatment, and a group with UC-MSCs or hydrogel alone to assess their individual effects on MIN-6 cell function.
4) You mentioned staining of live and dead cells to assess cell survival. Please provide more details about the staining technique used and the quantification of cell viability. Additionally, how were UC-MSCs and MIN-6 cells co-cultured, and what was the interaction between them within the hydrogel?
5) While you mention that UC-MSCs + hydrogel reduced oxidative stress, inflammation, and promoted β-cell proliferation, it would be helpful to elaborate on the underlying mechanisms. Were any specific signaling pathways or factors investigated to explain these effects?
6) Considering the potential application of UC-MSCs + hydrogel for diabetes treatment, were there any discussions or implications mentioned regarding future clinical translation? What are the challenges and limitations that need to be addressed before moving towards human studies?
7) You briefly compare UC-MSCs + hydrogel with existing non-hydrogel-encapsulated stem cell approaches for diabetes treatment. Could you provide a more comprehensive discussion on the advantages and limitations of the UC-MSCs + hydrogel approach, highlighting its unique contributions to the field?
Author Response
Reviewer 3
Comments and Suggestions for Authors
Dear Authors,
I have completed the review of your manuscript titled "Repair effect of UC-MSCs embedded in hydrogel on MIN-6 cells injured by STZ". However, I would like to recommend some revisions before further consideration.
Comments 1: What are the key findings of this study that contribute to the current understanding of diabetes treatment? How does this research advance the field and add to the existing knowledge?
Response: This study employed an innovative co-culture model that integrates umbilical cord mesenchymal stem cells (UC-MSCs) and STZ-injured MIN-6 cells in a composite hydrogel to simulate the UC-MSCs repair process of pancreatic β-cells under diabetic conditions. The combination of UC-MSCs and hydrogel was demonstrated to enhance the repair effects on the damaged MIN-6 cells compared to the UC-MSCs alone. The result displayed the potential of UC-MSCs integrated with hydrogel to facilitate the repair of pancreatic β-cells in diabetes therapy and offers new research directions for future treatment strategies for diabetes.
We aim to construct an implantable and minimally invasive tissue engineering material which is more stable, non-toxic and intelligent hydrogel material. The novel hydrogel materials design was expected to enhance stem cell therapies by providing an optimized microenvironment that supported cell survival, proliferation and differentiation, ultimately contributed to improve the treatment for T2DM patients.
Comments 2: Please provide more information on the statistical methods used to compare the different experimental groups and determine the significance of the results. Were the statistical tests appropriate for the data analysis?
Response: Thank you for the careful question. Data were presented as mean ± standard error of the mean (SEM). Inter-group comparisons were performed using one-way analysis of variance (ANOVA), followed by Tukey's honestly significant difference (HSD) post-hoc test. The asterisks (*) indicated the difference from the control group (*p < 0.05, **p < 0.01, ***p < 0.001).
Comments 3: It would be helpful to have a group with STZ-induced injury but without any repair treatment, and a group with UC-MSCs or hydrogel alone to assess their individual effects on MIN-6 cell function.
Response: Thank you for the valuable question. In this paper, the T2DM group represents the MIN-6 cells with STZ-induced injury but without any repair intervention. To demonstrate the repair effect of the encapsulation of stem cells in the composite hydrogel on the damaged MIN-6 cells, we have established the groups using UC-MSCs alone and a combination of UC-MSCs + hydrogel for comparative analysis, respectively.
Comments 4: You mentioned staining of live and dead cells to assess cell survival. Please provide more details about the staining technique used and the quantification of cell viability. Additionally, how were UC-MSCs and MIN-6 cells co-cultured, and what was the interaction between them within the hydrogel?
Response: Thank you for the valuable question. We used the Calcein AM, which is an excellent fluorescent dye for live cell labeling, capable of readily penetrating the cell membrane of viable cells to generate a green, fluorescent substance with a strong fluorescence signal. In contrast, for dead cells with compromised cell membranes, we used the propidium iodide (PI), which can enter the cell, bind to nucleic acids, and produce a bright red fluorescent signal indicative of dead cells. Quantitative assessment of cell viability was performed using the image processing software Image J for the quantitative analysis of fluorescence intensity. The co-culture procedure of UC-MSCs with MIN-6 cells was as follows: First, prepare the composite hydrogel precursor solution at 25°C, mix the counted UC-MSCs with the precursor solution, then allow the gel precursor solution to form a gel at 37°C. After the gel has formed, seed the MIN-6 cells onto the surface of the gel. Finally, culture with the addition of growth medium. We have added the contents in the revised manuscript.
Comments 5: While you mention that UC-MSCs + hydrogel reduced oxidative stress, inflammation, and promoted β-cell proliferation, it would be helpful to elaborate on the underlying mechanisms. Were any specific signaling pathways or factors investigated to explain these effects?
Response: Thank you for the valuable question. Umbilical cord mesenchymal stem cells (UC-MSCs) may be involved in the treatment of type 2 diabetes mellitus (T2DM), particularly in alleviating β-cell inflammation and oxidative stress, through the following pathways:
Notch Pathway [Front. Immuno. 11 (2020) 1391]: Notch signaling plays a crucial role in cell fate determination and differentiation. It may be involved in the protective effects of UC-MSCs on β-cells.
NF-κB and MAPK Signaling Pathways [Int. Endod. J. 55 (2022) 517-530]: Inhibition of inflammatory responses through the modulation of NF-κB and MAPK signaling pathways can significantly reduce the release of inflammatory cytokines.
TLR4/NF-kB [Tissue Cell, 66 (2020) 101382]: Improvement of organ damage caused by T2DM by regulating the TLR4/NF-kB inflammatory pathway and oxidative stress.
BDNF/Trkb Pathway [Cell Stress Chaperon. 28 (2023) 1041-1051]: Protection of damaged cells from apoptosis by modulating the BDNF/Trkb pathway.
ERK1/2 Pathway [Mat Sci Eng C, 120 (2020) 11671]: Promotion of cell proliferation, migration, and angiogenesis by upregulating VEGF levels through the ERK1/2 pathway, and acceleration of the healing process in diabetic tissues.
AKT Pathway [Stem Cell Res Ther, 13 (2022) 164]: Activation of the AKT pathway in macrophages leads to the upregulation of Arg1, a molecule associated with M2 macrophages, causing a shift towards an M2 phenotype. This is consistent with our previous findings that UC-MSCs can induce a transition of MI macrophages towards an M2 phenotype.
Comments 6: Considering the potential application of UC-MSCs + hydrogel for diabetes treatment, were there any discussions or implications mentioned regarding future clinical translation? What are the challenges and limitations that need to be addressed before moving towards human studies?
Response: Based on the results in our study, the challenges and limitations that need to be addressed before transitioning to clinical research are as follows. (1)The signal communication between stem cells and damaged MIN-6 cells in the composite hydrogel should be further investigated. The effect of the composite hydrogel on the directed differentiation of UC-MSCs also should be further explored. (2) The large-scale preparation of UC-MSCs in the composite hydrogel for application should be further researched, in which to establish a standardized detection system to ensure the tracking of cellular functional states, karyotype testing, and monitoring of relevant factors during the treatment process would be indispensable.
Comments 7: You briefly compare UC-MSCs + hydrogel with existing non-hydrogel-encapsulated stem cell approaches for diabetes treatment. Could you provide a more comprehensive discussion on the advantages and limitations of the UC-MSCs + hydrogel approach, highlighting its unique contributions to the field?
Response: Thank you for the valuable question. We have added the contents in conclusion of the revised manuscript.

Round 2
Reviewer 1 Report
Comments and Suggestions for Authors
We appreciate the author's efforts to improve the manuscript. Now this article can be accepted in its present form.